Ultra-performance liquid chromatography-mass spectrometry for precise fatty acid profiling of oilseed crops

Chernova Alina alin.chernova@gmail.com 1
Mazin Pavel 1 2 3
Goryunova Svetlana 1 4
Goryunov Denis 1 5
Demurin Yakov 6
Gorlova Lyudmila 6
Vanyushkina Anna 1
Mair Waltraud 1
Anikanov Nikolai 1
Yushina Ekaterina 1 7
Pavlova Anna 1
Martynova Elena 1 4
Garkusha Sergei 8
Mukhina Zhanna 8
Savenko Elena 8
Khaitovich Philipp 1
1 Center of Life Sciences, Skolkovo Institute of Science and Technology , Moscow , Russia
2 Faculty of Computer Science, National Research University Higher School of Economics , Moscow , Russia
3 Institute for Information Transmission Problems (Kharkevich Institute), Russian Academy of Sciences , Moscow , Russia
4 Institute of General Genetics, Russian Academy of Sciences , Moscow , Russia
5 Belozersky Institute of Physico-Chemical Biology, Lomonosov Moscow State University , Moscow , Russia
6 Pustovoit All-Russia Research Institute of Oil Crops , Krasnodar , Russia
7 Pirogov Russian National Research Medical University , Moscow , Russia
8 All-Russia Rice Research Institute , Krasnodar , Russia
Feussner Ivo
Electronic publication date: 2019 Mar 6
Publication date: 2019
Volume: 7
Electronic Location ID: e6547
Received 2018 Oct 30; Accepted 2019 Jan 31
Copyright: ©2019 Chernova et al.
Copyright year: 2019
Copyright holder: Chernova et al.
License: This is an open access article distributed under the terms of the Creative Commons Attribution License, which permits unrestricted use, distribution, reproduction and adaptation in any medium and for any purpose provided that it is properly attributed. For attribution, the original author(s), title, publication source (PeerJ) and either DOI or URL of the article must be cited.
License URL: https://creativecommons.org/licenses/by/4.0/

Keywords: Ultra-performance liquid chromatography-mass spectrometry, Oil crops, Rapeseed, Fatty acids, Breeding, Gas chromatography-flame ionization detection, Sunflower

Funding: Ministry of Science and Higher Education of the Russian Federation 14.609.21.0099 This research was funded by the Ministry of Science and Higher Education of the Russian Federation (Grant No. 14.609.21.0099, Identification No. RFMEFI609l6X0099). The funders had no role in study design, data collection and analysis, decision to publish, or preparation of the manuscript.

==============================
Oilseed crops are one of the most important sources of vegetable oils for food and industry. Nutritional and technical properties of vegetable oil are primarily determined by its fatty acid (FA) composition. The content and composition of FAs in plants are commonly determined using gas chromatography-mass spectrometry (GS-MS) or gas chromatography-flame ionization detection (GC-FID) techniques. In the present work, we applied ultra-performance liquid chromatography-mass spectrometry (UPLC-MS) technique to FA profiling of sunflower and rapeseed seeds and compared this method with the GC-FID technique. GC-FID detected 11 FAs in sunflower and 13 FAs in rapeseed, while UPLC-MS appeared to be more sensitive, detecting about 2.5 times higher numbers of FAs in both plants. In addition to even-chain FAs, UPLC-MS was able to detect odd-chain FAs. The longest FA detected using GC-FID was an FA with 24 carbon atoms, whereas UPLC-MS could reveal the presence of longer FAs with the tails of up to 28 carbon atoms. Based on our results, we may conclude that UPLC-MS has great potential to be used for the assessment of FA profiles of oil crops.

Introduction

Vegetable oils have been used by humans since ancient times. The main components of the vegetable oil are triglycerides (about 95%) (Thomas, Matthäus & Fiebig, 2015). They are composed of three fatty acids (FAs) attached to glycerol by ester bonds. FA profile is an important characteristic of the vegetable oil (Zhang et al., 2014). The analysis of FA content is one of the topical issues in plant metabolomics given the importance of deciphering lipid biosynthesis pathways in plants. Oilseed crops are one of the most important sources of vegetable oils for food and industry. Sunflower and rapeseed take the positions four and three in the global production of vegetable oil after palm and soybean (Rauf et al., 2017). In oilseed crops, the identification of FAs appears to be of special importance since nutritional and technical properties of the oil extracted from these crops are primarily determined by its FAs composition. For instance, unsaturated fatty acids contribute to decreasing cholesterol levels in blood thus reducing the risk of heart diseases (Grundy, 1986).

Apart from their essential role in nutrition, sunflower and rapeseed oils have a number of industrial applications: they can be used as basic components in polymer synthesis, serve as the source of biofuel, or be used as emulsifiers or lubricants (Dimitrijević et al., 2017). Production of selection hybrids with changed oil properties is one of the key objectives of oilseed crop breeding (Jocić, Miladinović & Kaya, 2015). Seed oil content and quality were subjected to selection at all times in the history of sunflower and rapeseed crop improvement (Chapman & Burke, 2012). One of the main directions in sunflower and rapeseed breeding is the selection for high oleic oil. Such oil is characterized by higher degree of oxidative stability, which makes it more suitable for frying, refining, and storage (Fuller, Diamond & Applewhite, 1967; Premnath et al., 2016). This phenomenon finds many applications in food, automotive, and textile industries.

Although sunflower and rapeseed breeding demonstrate certain similar traits, there exists significant differences in the directions of the selection process in these two crops. The oil from rapeseed naturally contains high quantities of erucic acid and glucosinolates. Due to this fact, the use of rapeseed oil in food industry was restricted because of its physiopathological effect on mammals (Borg, 1975). These limitations were overcome in the 1960s by the production of rapeseed whose oil was free from erucic acid (Stefansson & Hougen, 1964).

The oilseed crop market constantly sets new trends in rapeseed and sunflower breeding, which stimulates plant breeders to develop new varieties with optimized FA content (Velasco & Fernández-martínez, 2002). Towards this end, it appears necessary to elaborate efficient protocols for precise and high-throughput FA profiling. Presently commonly used techniques to measure FAs in plants are gas chromatography-mass spectrometry (GS-MS) or gas chromatography-flame ionization detection (GC-FID) methods (Li-Beisson et al., 2010). Prior to GC analysis, FAs are usually converted into the corresponding methyl ester derivatives (FAMEs) by methylation and trans-esterification (Liu, 1994).

By combining the advantages of fast high-resolution chromatographic separation with the high-sensitivity, selectivity, and specificity of mass spectrometric detection, UPLC-MS became the technique of choice for a wide range of applications (Pitt, 2009; Hummel et al., 2011; Want et al., 2013). Bromke et al. (2013) and Bromke et al. (2015) suggested using liquid chromatography high–resolution mass spectrometry (UPLC-MS) for FA profiling in Arabidopsis thaliana and diatoms.

In light of all that was mentioned above, the aim of the present study was to test the use of ultra-performance liquid chromatography-mass spectrometry (UPLC-MS) coupled with lipid extraction using methyl tert-butyl ether for precise analysis of FA composition of oilseed crops (sunflower and rapeseed).

Materials & Methods

Plant material

Seeds from 50 sunflower (Helianthus annuus) and 50 rapeseed (Brassica napus) lines (Table S1) from the Pustovoit All-Russia Research Institute of Oil Crops Collection (Russia, Krasnodar) were used in the study.

Reagents

The following reagents were used: Methanol LC-MS (Scharlau, Barcelona, Spain), Methyl tert-butyl ether HPLC grade (Scharlau, Barcelona, Spain), Chloroform HPLC grade (Fisher Scientific, Waltham, MA, USA), Heptane LC/MS grade (Honeywell Fluka, Morris Plains, NJ, USA), Water UHPLC-MS grade (Scharlau, Barcelona, Spain), Potassium hydroxide solution 45% in water (Sigma Aldrich, USA), NaCl USP grade (Helicon, Moscow, Russia), HCl 37% (PanReac AppliChem, Omaha, NE, USA), Acetonitrile LC/MS grade (Fisher Scientific, Waltham, MA, USA), Isopropanol LC-MS grade (Honeywell Fluka, USA), Ammonium acetate (Honeywell Fluka, Morris Plains, NJ, USA), Formic acid 98%–100% LC-MS grade (LiChropur Merck Millipore, Burlington, MA, USA), Acetic acid Optima, LC-MS grade (Fisher Scientific, Waltham, MA, USA), and FA standards (Oleic acid-13C18 (Sigma Aldrich, St. Louis, MO, USA), Palmitic acid-13C16 (Sigma Aldrich, St. Louis, MO, USA), and Stearic acid-13C18 (Sigma Aldrich, St. Louis, MO, USA)).

GC-FID analysis

A total of 4–5 g of seeds were mixed together and homogenized and 0.5 g was taken for fatty acid extraction with 4 ml of hexane.

To obtain the methyl esters of fatty acids, 2–3 ml of homogenized seed-hexane mixture was transferred into the new tube and 0.1 ml of sodium methylate was added, and mixed intensively for two minutes. The tube content was further transferred onto the paper filter with Na2SO4 on the bottom. Obtained filtrate was then placed into the DAG-2M automatic dispenser tube.

GC-FID analysis was carried out using the < <Chromateck-Crystall 5000 > > GC chromatograph with the DAG-2M automatic dispenser. GC separation was performed in a SolGelWax column with the dimensions of 30 m × 0.25 mm × 0.5 µm; gas mobile phase - helium; speed - 25 cm/sec; temperature range - 185–230 °C.

UPLC-MS analysis

For lipid extraction, 10 mg (for each line) of sunflower (1 sample - 1 seed) and rapeseed (1 sample - several seeds) seeds were homogenized using six 2.8 mm zirconium oxide beads (Bertin Instruments, Montigny-le-Bretonneux, France) in the Precellys Evolution homogenizer (Bertin Instruments, Montigny-le-Bretonneux, France) coupled with Cryolis filled with dry ice at the temperature not exceeding 10 °C. Homogenization parameters were as follows: 6,800 rpm, 3* 20 s, pause 30 s. 400 µL of methanol/methyl tert-butyl ether mixture (1: 3 v: v) was added prior to homogenization. After homogenization, another 400 µl of methanol/methyl tert-butyl ether mixture was added, and the sample was mixed by vortexing. After sonication for 10 min in the ice-cooled ultrasonic bath and incubation at 4 °C for 30 min with shaking, the sample was transferred into the new 1.5 ml Eppendorf tube, and 560 ml of water/methanol mixture (3: 1 v: v) was added. After the addition of water/methanol the sample was mixed by vortexing for 10 min and centrifuged for 10 min at 4 °C at 12,700 rpm. This leads to the separation of two phases: the lipophilic phase and the polar phase. The upper, lipophilic, phase was collected and vacuum dried in Concentrator plus (Eppendorf) for 1.5 h at 30 °C and the pellet was stored at −80 °C before FA measurement. This protocol is based on the protocol described by Giavalisco et al. (2011) (Giavalisco et al., 2011). For FA quantification, three isotopically labeled internal FA standards were added to the extraction mixture (3 µg of each FA per sample).

For the analysis of FAs, the extracts obtained at the previous step were hydrolyzed using the protocol adopted from Bromke (Bromke et al., 2015). Lipid extracts were resuspended in 200 µl of the mixture of methanol and 6% KOH (4: 1 v: v). The tubes were incubated for 2 h at 60 °C with continuous shaking (1,800 rpm). After cooling to room temperature, 100 µl of saturated NaCl solution was added. The reaction mixture was acidified by the addition of 50 µl of 29% HCl. Tubes were thoroughly vortexed and spun for 30 s at full speed using a table centrifuge. FAs were extracted with 200 µl of chloroform/heptane mixture (1: 4 v: v). After mixing by vortexing and 15 s centrifugation, the organic phase was collected. Extraction with chloroform /heptane mixture was repeated the second time and the collected FA-containing organic phases were combined. The extract was washed by the addition of 200 µl of water followed by short vortexing and 15 s centrifugation at 12,700 rpm, which resulted in the separation of two phases. Finally, the organic upper phase was collected, dried under vacuum conditions in Concentrator plus (Eppendorf) (30 min, 30 °C) and stored at −80 °C before FA measurement.

To prepare samples for the injection into the UPLC-MS system, the dried extracts were first resuspended in acetonitrile/isopropanol mixture (70: 30), vortexed for 10 s, and incubated for 10 min at 4 °C with shaking, followed by sonication for 10 min on ice and centrifugation for 7 min at 4 °C at the speed 12,700 rpm. After the completion of these procedures, final dilutions with acetonitrile/isopropanol (70: 30) were prepared in MS vials.

Samples were processed using mass spectrometry (UPLC-MS) coupled with reversed phase ultra-performance liquid chromatography (ACQUITY UPLC System; Waters, USA) on Q-TOF (Quadrupole-Time-of-Flight) Maxis Impact II mass spectrometer (Bruker Daltonics, Wien, Austria). Parameters for analysis were set according to the negative ion mode with the spectra acquired over a mass range from m/z 50 to 1,200. The optimum values of the ESI-MS parameters were as follows: capillary voltage, + 4.0 kV; drying gas temperature, 200 °C; drying gas flow, 6.0 l/min; nebulizing gas pressure, 2 bar.

UPLC separation was carried out using the C8 Acquity Beh column (2.1 mm ×  100 mm, 1.7- µm particle size; Waters) and the Acquity BEH C8 1.7 µm Vanguard precolumn (Waters) at 60 °C. For UPLC gradient the mobile phases consisted of two solvents. Two different solvent systems and gradients were tested.

1. Solvent A: 1% of 1 M NH4Ac solution and 0.1% formic acid in water; and solvent B: acetonitrile/isopropanol (7: 3, 1% of 1M NH4Ac solution and 0.1% formic acid), with an injection volume of 3 µl. The following gradient profile was applied: 55% B, 1 min; linear gradient from 55% B to 80% B, 3 min; linear gradient from 80% B to 85% B, 8 min; linear gradient from 85% B to 100% B, 3 min. After washing the column for 4 min 50 s with 100% B, the proportion of buffer B in the mixture was set back to 55%, and the column was re-equilibrated for 4 min 10 s (24.5 min total run time), with the mobile phase flow rate of 400 µl/min.

2. Solvent A: 1% of 1M NH4Ac solution and 0.1% acetic acid in water; and solvent B, acetonitrile/isopropanol (7: 3, 1% of 1M NH4Ac solution and 0.1% acetic acid), with an injection volume of 3 µl. The following gradient profile was applied: 55% B, 50 s; linear gradient from 55% B to 75% B, 1 min; linear gradient from 75% B to 89% B, 5 min; linear gradient from 89% B to 100% B, 1 min 10 s. After washing the column for 2 min with 100% B, the proportion of buffer B in the mixture was set back to 55%, and the column was re-equilibrated for 1 min 50 s (11.5 min total run time), with the mobile phase flow rate of 400 µl/min.

The sample final dilution was 5 (UPLC-5) and 400 times (UPLC-400) for the buffer systems 1 and 2, respectively.

Method validation procedure

Validation procedure included the evaluation of linearity, repeatability, reproducibility, and calculation of the limit of detection (LOD) and the limit of quantification (LOQ). Linearity was evaluated by building seven-point calibration curves (six replicates) in pure solvents. Solvent (acetonitrile/isopropanol, 70/30) was spiked with a standard mixture of isotopically labeled fatty acids (Oleic acid-13C18, Palmitic acid-13C16, and Stearic acid-13C18) in order to obtain all standards in the concentrations of 5, 10, 50, 100, 500, 1,000, and 2,500 ng/ml. The mean correlation coefficients (n = 6) of the calibration curves were 0.997 or higher for the target analytes, indicating good linearity in the selected concentration range (Tables 1A and 1B).

The limit of detection (lod) and the limit of quantification (LOQ) were determined from spiked samples, as the minimum detectable concentration of analyte with a signal-to-noise ratio of 3 and 10, respectively.

For repeatability evaluation, MS technical repeats were performed. The results are presented in the Table S2.

Data processing analysis

GC-FID: FAs detection was performed based on retention times using FA methyl ester standards (Fluke). The percentage of each FA was calculated based on the peak area using GC software.

Table 1 Linearity, LOD, LOQ, correlation coefficients (R) of the target compounds for LC-MS method validation (A-ammonium acetate with acetic acid addition, B- ammonium acetate with formic acid addition).

Compound	Equation	Linearity, ng/ml	R	LOD (ng*ml−1)	LOQ (ng*ml−1)	
A	
Oleic acid-13C18	y = 2424.2x − 4768.3	50–1,000	0.999	15	50	
Palmitic acid-13C16	y = 5908.5x − 38971	10–2,500	0.997	5	10	
Stearic acid-13C18	y = 2715.3x − 62204	50–1,000	0.999	15	50	
B	
Oleic acid-13C18	y = 273.87x − 12117	100–2,500	1	40	100	
Palmitic acid-13C16	y = 354.89x − 14928	50–2,500	0.999	20	50	
Stearic acid-13C18	y = 190.76x − 8817.5	100–2,500	0.996	40	100	

Xcms package (Smith et al., 2006) was used for UPLC-MS data processing. Optimal parameters were chosen with the aid of the IPO package (Libiseller et al., 2015). Peak intensities were obtained using the peakTable function implemented in the xcms package with method=‘maxint’. As a result, set of peaks characterized by the retention time, mass divided by charge, and intensity for each sample was obtained. Isotopically labeled internal FA standards were used for retention time correction and alignment.

To identify peaks that correspond to FAs, all possible chemical formulas that correspond to FAs with the chain length varying between 10 and 28 carbon atoms (n) and with the number of double bonds varying from 0 to 6 (k) defined as CnH2n−2kO2 were generated. Since the method does not distinguish between isomeric FAs, in what follows, the term ‘FA’ will be used to denote the groups of isomers (example of isomeric pattern presented on Fig. S1) defined by certain n and k, for example, FA18:1 denote all fatty acids with 18 carbons and single double bond. The masses of all these FAs in deprotonated state were calculated and all peaks matched masses within 20 ppm (part per million, for two masses m1 and m2 ppm = abs(m1 − m2)∕max(m1, m2)∗106) were searched for.

Then peaks were manually selected based on the net-like patterns as described in [15]. Briefly, it was observed that the extension of the FA length increases retention time, while the addition of the double bond reduces retention time, thus FA peaks form net-like pattern.

For the sunflower LC-5 FA18:3 two peaks with the ten-second difference between their retention times were obtained. Since the intensities of both these peaks exhibited high Pearson correlation with the FA18:3 intensity in LC-400 data the sum of these peaks was used in the analysis. Intensities of all detected FAs were at least three orders of magnitude higher in plant samples than in blank samples. To obtain FA mass fractions, the intensities of all FAs were divided by the sum of intensities of FAs identified using all three methods (GC-FID, UPLC-5, and UPLC-400) and multiplied by 100.

Statistical analysis and data visualization was carried out using R (R Core Team, 2013).

Results

In the present work, we compared the application of GC-FID coupled with hexane extraction and two UPLC-MS based approaches: 1:5 dilution in the buffer system with formic acid (UPLC-5) and 1:400 dilution in the buffer system with acetic acid (UPLC-400) for FA profiling in sunflower and rapeseed, both using MTBE extraction.

GC-FID data

Fifty lines of either of the two oilseed crops were analyzed using GC-FID. As a result we have detected 11 and 13 FAs in sunflower and rapeseed, respectively (Figs. 1C and 2C, Table S3). The lists of FAs which were detected in the two crops differed by minor FAs, while C16:0, C16:1, C18:0, C18:1, C18:2, C18:3, C20:0, C20:1, C22:0, and C24:0, were common for both of them. FA C20:2, C22:1, and C24:1 were detected in rapeseed only, while C14:1 FAs we found in sunflower but were absent in rapeseed (Figs. 1C and 2C).

Figure 1 Dependency between the mass fractions of FAs in sunflower (calculated for 11 FAs detected by GC-FID) estimated using different methods.

GC-FID vs UPLC-5 and GC-FID vs UPLC-400 are shown in (A) and (B), respectively. Each dot corresponds to a given FA (shown by color) in a given sample (C) Mean mass fractions (relative to the total intensity of 11 FAs detected by GC-FID) for all FAs determined using the three techniques. FAs which were not detected by the indicated method are shown by white rectangles. FAs are ordered according to their intensities as obtained by UPLC-5.

Figure 2 Dependency between the mass fractions of FAs in rapeseed (calculated for 11 FAs detected by GC-FID) estimated using different methods.

GC-FID vs UPLC-5 and GC-FID vs UPLC-400 are shown in (A) and (B) respectively. Each dot corresponds to a given FA (shown by color) in a given sample (C) Mean mass fractions (relative to the total intensity of 11 FAs detected by GC-FID) for all FAs determined using the three techniques. FAs which were not detected by the indicated method are shown by white rectangles. FAs are ordered according to their intensities as obtained by UPLC-5.

In rapeseed oil, the most abundant FAs were C16:0, C18:0, C18:1, C18:2, C18:3, and C20:1 which together made for 98.6% of all FAs. In sunflower seed oil, the most abundant FAs were C16:0, C18:0, C18:1, C18:2, C22:0, which constituted 98.9% of all FAs. In both species, two FAs (18:2 and 18:1) constitute more than 80% of the total FA content. However, in rapeseed, 18:1 is three times more abundant than 18:2, while in sunflower it is 18:2 which dominates (Figs. 1C and 2C).

The relative abundance of FAs was demonstrated to vary between samples (Table S3).

UPLC-MS data

For UPLC-MS fatty acids profiling we used two different buffer systems: the UPLC buffer with formic acid and the buffer with acetic acid. The former buffer is the most common buffer used in UPLC-MS experiments. In the latter buffer, ionization is known to be more effective since acetic acid is a weaker acid, so lower sample amount, and, therefore, higher sample dilution is required. For this reason for the system with acetic acid we used higher dilution (1:400), and for the system with formic acid we used lower sample dilution (1:500).

Hydrolyzed extracts of all lines of both sunflower and rapeseed were analyzed using both UPLC-MS-based methods.

At low dilution (UPLC-5), we are able to detect 29 FAs for sunflower and 35 FAs for rapeseed. At high dilution (UPLC-400), 25 and 31 FAs were detected for sunflower and rapeseed, respectively (Figs. 1C and 2C, Table S2). Hence, the use of lower dilution rates proved to be a more efficient approach which allowed to reveal the highest number of FAs. However, in this case it becomes impossible to perform the relative quantification of major FAs: 18:1 and 18:2, because they cause detector saturation (Figs. 3A, 3B). Apart from even-chain FAs, UPLC-MS also revealed odd-chain FAs (for example 17:0, 17:1, and 17:2). FAs with 18-carbon chain (stearic, oleic, and linoleic) were the most abundant. A net-like pattern on the M/z-RT plots can be observed (Figs. 4A, 4B and 5A, 5B).

Figure 3 LC chromatograms obtained for sunflower.

(A) UPLC-5. (B). UPLC-400. Chromatographic peaks corresponding to 18:1 and 18:2 FAs stay out of dynamic range in (A). Higher dilution rates aid in resolving this problem (B).

Figure 4 Comparison of different techniques for quantitative assessment of FAs in sunflower.

(A, B) Retention time (x-axis, RT)–m/z (y-axis) scatter plots for UPLC-5 (A) and UPLC-400 (B). Each dot corresponds to the individual FA, mean log-intensity is indicated by the dot size. Red color text indicates chain length and number of double bounds. (C) Dependency between the mole fractions estimated using different methods, UPLC-5 vs UPLC-400. Pearson correlation was calculated between logs of mole fraction.

Figure 5 Comparison of different techniques for FA quantification in rapeseed.

(A, B) Retention time (x-axis, RT)–m/z (y-axis) scatter plots for UPLC-5 (A) and UPLC-400 (B). Each dot corresponds to the individual FA, mean log-intensity is indicated by the dot size. Red color text indicates chain length and number of double bounds. (C) Dependency between the mass fractions estimated using different methods, UPLC-5 vs UPLC-400. Each dot corresponds to the individual sample.

FA content show a big variability between the lines (Table S2 , Fig. S2).

Results of both UPLC-MS methods show strong correlation: Spearman rho = 0.927 and 0,854 for sunflower and rapeseed, respectively (Figs. 4C, 5C).

Comparison of UPLC-MS with GC-FID

Results of UPLC-MS were compared with the results obtained using GC-FID. All 11 FAs detected in sunflower and 13 FAs detected in rapeseed were also identified using UPLC-MS. In the case of both plants, UPLC-MS proved to be more sensitive and detected about 2.5 times higher number of FAs than GC-FID (Figs. 1C, 2C). The majority of the minor FAs (mass fraction below 0.5%) are missed by GC-FID. The longest FA detected by GC-FID was the FA with 24-carbon chain, whereas UPLC-MS provided insight into the changes in longer FAs with the chains containing up to 28 carbon atoms.

As it can be seen in Figs. 1A, 1B and 2A, 2B, although the relative amounts of FAs measured by UPLC-MS and GC-FID are different, there exist significant correlation between them. Spearman rho = 0.908 and 0,918 for sunflower GC-FID and UPLC-5 and GC-FID and UPLC-400, respectively. Spearman rho = 0.979 and 0,947 for rapeseed GC-FID and UPLC-5 and GC-FID and UPLC-400, respectively.

Discussion

FA composition of seed oil from 50 sunflower and 50 rapeseed lines was analyzed by GC-FID, which is traditionally used to measure FAs in plant oils, and two UPLC-MS-based approaches (1:5 dilution in the buffer system with formic acid (UPLC-5) and 1:400 dilution in the buffer system with acetic acid (UPLC-400)). GC-FID technique allowed to detect 11 and 13 FAs in sunflower and rapeseed, respectively, all of them representing even-chain FAs.

GC-FID based FAs abundances in sunflower and rapeseed are in good agreement with those obtained in previous investigations (Alpaslan & Gündüz, 2000; Pleite, Martínez-Force & Garcés, 2006; Demurin & Borisenko, 2011; Bocianowski, Mikołajczyk & Bartkowiak-Broda, 2012; Sharafi et al., 2015).

UPLC-MS is a more sensitive technique compared with GC-FID, so it was not surprising that we detected a considerable number of additional components in the FA profiles obtained using this method. This result corresponds well with the data obtained by Bromke et al. (2015) for Arabidopsis thaliana tissues.

In total, about 29 and 35 FAs were detected in sunflower and rapeseed samples, respectively, by UPLC-MS. It is worth noting that utilizing the UPLC-5 approach we were able to identify significantly more FAs in both crops compared with the UPLC-400 approach. However with the 1:5 dilution implemented in this approach it was impossible to perform the relative quantification of the two most abundant FAs, 18:1 and 18:2, due to detector saturation. In view of this, we suggest that the UPLC-400 approach is more suitable for FA profiling of oil crop samples.

We have demonstrated in our study that UPLC-MS method is appropriate for detection of long FAs both in sunflower and rapeseed. The longest FA detected by GC-FID method was the FA with 24-carbon chain, whereas UPLC-MS revealed FAs with the tails up to 28 carbon atoms long. Generally, about half of the FAs identified by UPLC-MS belong to Very-long-chain fatty acids (VLCFA, fatty acids with the chain length of at least 20 carbon atoms). According to the previous investigations, VLCFAs are mainly located in the cuticular wax layer deposited at the surface of plant aerial organs; they form part of triacylglycerides of seed oil and sphingolipids and are essential for many aspects of plant development and apparently play role as signal molecules governing both biotic and abiotic stress (Roudier et al., 2010; Bach & Faure, 2010; De Bigault Du Granrut & Cacas, 2016).

Apart from even-chain FAs, odd-chain fatty acids were also detected by UPLC-MS. The latter are present in the oil extracted from the analyzed plants only in minor quantities and selection for these FAs is not currently included in breeding programs. However, beneficial effects of these compounds on human health have been recently demonstrated. For example, pentadecenoic and heptadecenoic acids contribute to reduced risks of developing multiple sclerosis and act as anti-inflammatory and edema-inhibiting agents (Degwert, Jacob & Steckel, 1998; Jenkins et al., 2017). Additionally, odd-chain fatty acids inhibit the development of certain plant pathogens and could be used as precursors for manufacturing agricultural and industrial chemicals (Fitton & Goa, 1991; Avis, Boulanger & Bélanger, 2000; Avis & Bélanger, 2001; Clausen, Coleman & Yang, 2010; Köckritz, Blumenstein & Martin, 2010). The chemical properties and potential biological activities of odd-chain fatty acids are continuously under investigation(Rezanka & Sigler, 2009). Due to the importance of odd-chain fatty acids efforts are being made to produce yeast strains with increased content of FAs of this kind (Park et al., 2018).

Therefore, the analysis of minor FA content in oil may be important to make a complete assessment of the functional and nutritional properties of the oil. Our results suggest that UPLC-MS has great potential as a precise tool for evaluation of the full FA profile in oilseed crops. It can be essential for the creation of vegetable oils with increased nutrition value and/or new technical characteristics and provide additional markers for agronomically important traits in plants.

Considering the advantages of UPLC-MS and its applicability for FA profiling in oilseed crops, we may observe that the overall result can be represented as a net-like pattern on the retention time M/z-RT plots (Figs. 4A, 4B and 5A, 5B). This makes the process of results interpretation and annotation easier compared to GC-FID results.

Another advantage of UPLC-MS is the possibility of analyzing thousands of samples per month and small amount of plant material needed for the analysis. This technique requires only 5–10 mg of plant material per extraction, while GC-FID requires high amounts of material. For plants with big seeds like sunflowers it is possible to take only a small part of the seed for the FA profiling analysis and germinate the rest of the seed and plant it, which allows exact assignment of phenotype to genotype in breeding programs.

It also worth mentioning that UPLC-MS technique involves no FA derivatization, which allows identifying more FAs compared to the conventional GC-FID technique. The results of the present study confirm this and they are in good correspondence with the data obtained by Bromke et al. (2015) for Arabidopsis.

It is important to highlight that certain differences between the results obtained by GC and UPLC-MS may be connected with the lipid extraction procedure. We used MTBE extraction which extract both TAGs and phospholipids, which means that some detected FAs may come from phospholipids, compared to GS where extraction was performed with hexane which normally extracts mostly TAGs.

Based on the results obtained in the present study, we conclude that UPLC-MS is a promising technique which may be used for FA composition analysis in oil crops as it is highly sensitive, scalable, and suitable for the individual seed analysis.

Currently, gas chromatography with mass spectrometric or flame ionization detection is the gold standard for quantitative assessment of FA composition of vegetable oils. Based on our results, UPLC-MS has great potential to be used in evaluation of FA composition of oil crops as highly sensitive, scalable, and suitable for the individual seed analysis technique. In our study, it has been shown that GS-MS/GS-FID could be substituted by UPLC-MS in a number of cases. However, additional comparative studies are required for UPLC-MS method to become a standard technique for evaluation of oil FA composition both in breeding programs and for industrial purposes.

Conclusions

In this work we compared the performance of hexane lipid extraction with GC-FID and MTBE lipid extraction with UPLC-MS when measuring FA profiles in 50 sunflower and 50 rapeseed samples. Based on the obtained results, we may conclude that UPLC-MS with MTBE extraction has great potential for the study of FA composition of oilseed crops, which may be used both to evaluate the oil properties for nutritional and industrial needs, to perform high-throughput analysis to assist oilcrop breeding, as well as to search for new potential valuable traits. The main advantages of this technique as it was demonstrated in this study is high-sensitivity, it which detects 2.5 times more FA species both in sunflower and rapeseed than the conventional GC-FID technique, including the minor fatty acids and odd-chain fatty acids, which are usually omitted. Additionally, the UPLC-MS technique is able to detect FAs with long carbon chains, including those with 28-carbon tails. The UPLC-MS technique doesn’t require FA derivatization which also adds to its high-sensitivity. It is worth mentioning, that by using this technique it appears possible to use small of starting material which makes it very useful in breeding programs since it appears possible to use only a small part of the seed for analysis and to germinate the remaining part to use it in breeding programs, for example for subsequent phenotyping. Taken together our findings suggest that UPLC-MS provides a deep insight into the oil FA content and may be applied for precise identification of FA profiles of oilseed crops, although further comparative studies using larger samplings with further improvements and optimization are required.

Supplemental Information

Table S1 List of samples used in the study

Each sample is indicated by Species name, winter or spring type (for rapeseed), line id number from the catalog and id number used in mass specrometry analysis.

Click here for additional data file.

Table S2 UPLC-MS data

(A) UPLC-MS data for sunflower and rapeseed. For each sample in the table the information about species, method of analysis (UPLC-5 and UPLC-400), mass spectrometry id number and the intensity of each FA are presented. (B): sunflower UPLC-5 statistics. For each FA mean value, standard deviation, minimum and maximum values and coefficient of variation are presented. (C): sunflower UPLC-400 statistics. For each FA mean value, standard deviation, minimum and maximum values and coefficient of variation are presented. (D): rapeseed UPLC-5 statistics. For each FA mean value, standard deviation, minimum and maximum values and coefficient of variation are presented. (E): rapeseed UPLC-400. For each FA mean value, standard deviation, minimum and maximum values and coefficient of variation are presented. (F): repeatability control: technical quality control samples (tQC). For each tQC the intensity of each FA are presented. For each FA mean value, standard deviation, minimum and maximum values and coefficient of variation are presented calculated from tQC samples.

Click here for additional data file.

Table S3 GC-FID data

(A): GC-FID data for sunflower and rapeseed. For each sample in the table the information about species, line id and GC data for each FA are presented. (B): rapeseed GC-FID statistics. For each FA mean value, standard deviation, minimum and maximum values and coefficient of variation are presented. (C): sunflower GC-FID statistics. For each FA mean value, standard deviation, minimum and maximum values and coefficient of variation are presented.

Click here for additional data file.

Figure S1 Example of isotopic pattern

A. Extracted ion chromatogram for 18:1 FA B. MS spectrum for 18:1 FA with 3 isotope peaks. X-axis -retention time in min, Y-axis -intensity in counts.

Click here for additional data file.

Figure S2 Variability between lines

The abundance of major FAs vary between sunflower samples measured by UPLC-5. Each colored line -chromatogram of 1 sample. X-axis -retention time in min, Y-axis -intensity in counts.

Click here for additional data file.

Additional Information and Declarations

Competing Interests

Author Contributions

Data Availability

The authors declare there are no competing interests.

Alina Chernova conceived and designed the experiments, performed the experiments, analyzed the data, prepared figures and/or tables, authored or reviewed drafts of the paper.

Pavel Mazin analyzed the data, prepared figures and/or tables, authored or reviewed drafts of the paper.

Svetlana Goryunova analyzed the data, authored or reviewed drafts of the paper, approved the final draft.

Denis Goryunov authored or reviewed drafts of the paper.

Yakov Demurin, Anna Vanyushkina and Waltraud Mair conceived and designed the experiments.

Lyudmila Gorlova, Nikolai Anikanov, Ekaterina Yushina, Sergei Garkusha and Elena Savenko performed the experiments.

Anna Pavlova contributed reagents/materials/analysis tools.

Elena Martynova authored or reviewed drafts of the paper, approved the final draft.

Zhanna Mukhina carried out project administration.

Philipp Khaitovich approved the final draft, supervision.

The following information was supplied regarding data availability:

The raw data are available in the Supplemental Files.

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
