# Peer review of "Ultra-performance liquid chromatography-mass spectrometry for precise fatty acid profiling of oilseed crops"

_PeerJ, doi:10.7717/peerj.6547_

## Round 0.1 · original submission · Major Revisions

· Academic Editor

Major Revisions

Please carefully revise your manuscript and consider the points raised by reviewers #2 and #3. In the revised manuscript you should provide experimental evidence and arguments why one could compare both methods although different workflows (extraction etc.) were used.

·

Basic reporting

The manuscript for "Ultra-Performance Liquid Chromatography Mass-Spectrometry for precise fatty acid profiling of oilseed crops" is well written and data has presented in an interesting manner. Please correct a few spelling mistakes in the manuscript as follows:

Line 127 Change eppendorf to Eppendorf
Line 142 Correct mixining to mixing
Line 143: Correct chlorophorm to chloroform
Line 183 Correct standarts to standards
Line 187 Change peak Table to peak table
Line 340, change sunflower to sunflowers

Experimental design

Well design experiment with the appropriate number of replicates for the analysis. However, some useful information is missing in the manuscript and it will be useful to have that information added. Please include the validity of the two LC-MS methodologies with linearity, repeatability and reproducibility of the data. In addition, include the lower level of detection (LOD) as well as lower level of quantification (LOQ) using standards for each fatty acid standard reported.

Validity of the findings

This is a novel approach of detecting fatty acids using LCMS and it will be useful to the research community in fatty acid analysis in plant and biomedical research.

Reviewer 2 ·

Basic reporting

The writing is mostly adequate, but some checking for subject-verb agreement would help, especially in the methods section.

Sufficient references and context are provided.

It’s good that all the original data is provided in the supplemental tables. However, it could be organized in fewer tables. For example, all the original data could be combined with Table 1, by labeling each column that follow the sample information with a method/metabolite and including the data for that method/metabolite for each sample.

Also I could not locate legends for the supplemental tables and figures.

Lines 192-193: “Since the method used do not allow to distinguish between isomeric FAs”-- Suggested rewording: “Since the method does not distinguish isomeric FAs”

Tables: “Procentage fraction” should be “percent”. Significant digits should reflect the precision of the measurements (i.e., excess digits appear to be present in GC data). “NA” should be defined.

Figures 3-5: Legends should describe the samples (lines).

Line 349: “which allows detecting 2.5 times higher amounts of FA” -- Consider: “which detects 2.5 times more FA species”

Line 142: mixining should be mixing

Line 143: chlorophorm should be chloroform

Line 183: standarts should be standards

Line 248: sperman rho should be Spearman.

Lines 183 and 215: GS should be GC? If so, you don’t need to add “chromatograph”.

Supplemental Table 1: Sheet label is not in English.

Experimental design

The article is within the scope of the journal. The research question asks how LC-MS with TOF detection in negative mode compares with GC as a method for fatty acid analysis. It demonstrates that LC-MS can detect more fatty acid species, but that the quantification is poor. It was a little disappointing that there wasn’t any attempt at absolute quantification (using internal standards) for either method and that there was apparently no attempt to describe the linearity of the LC-MS detected response as a function of the amount analyzed. Additionally, the fact that there are two methods, one with one chromatography solvent system and concentrated samples and the other with a different solvent system and dilute samples makes it difficult to assess the effect of the solvent system and the sample concentration independently. Was there any advantage/disadvantage of one solvent system over the other?

Line 133-134, describing the LC method: “For FA quantification, three isotopically labeled internal FA standards were added to the extraction mixture (3 μg of each FA per sample).” These standards do not appear to be used in any calculations. Only calculating percentages is described (lines 206-208).

If the authors wanted to make a great LC-MS method for fatty acids, it would be possible to determine the response factor for each fatty acid by comparison to the GC data, ideally over a range of concentrations through which both methods are linear.

Validity of the findings

The authors should consider in their calculations that GC-FID response is approximately proportional to the mass of the fatty acid methyl esters, and, thus, to calculate mole percent, the molecular weights of the methyl esters should be included in the calculations. On the other hand, MS response is theoretically proportional to moles (if all species ionize equally, though this doesn’t seem to be the case) and mole percent can be calculated directly from intensities. Putting values from GC and LC-MS both in mole percent would be appropriate.

Another thing that the authors should/could consider is that, because the m/z of the fatty acids vary quite a lot, the fraction of the intensity in the major isotopologue also varies. For example, for 14:0, the lowest m/z peak at 227 for [M-H]- accounts for ~ 85% of the total intensity, whereas for 28:0, the lowest m/z peak at 423 accounts for only ~73% of the total intensity, so it would be appropriate to divide the intensity associated with the main isotope by the intensity of that peak over of the total intensity of all the isotopes (i.e., 0.85 or 0.73). This would alter the percentages a bit.

What is the rationale for using Spearman’s correlation coefficient vs. Pearson’s for method comparison? It seems like you want to determine if there is a linear relationship which might be best accomplished with Pearson’s correlation.

In Supplemental table 3, the authors calculate the coefficient of variation across the samples. It would be nice to see the coefficient of variation calculated on the same sample run at intervals throughout the analysis. That would give information about the repeatability of the analysis.

Additional comments

Figure 1: The panels for A and B seem to be reversed in relation to the description. This figure would be better if the peaks were labeled with the fatty acids that they represent. “Dynamical” should be deleted or replaced with “dynamic”.

Line 68: “significant differences in the directions of the selection process in these two crops”: No mention is then made of anything about sunflower selection.

Lines 199-201: The mention of “net-like” patterns here does not seem to make this any clearer than just indicating the effect of chain length and double bond number on the retention time.

Line 157: It would be helpful to indicate that the MS scan is with the TOF, rather than the quadrupole.

Line 322: “while GC-FID requires at least 4-5g” This seems like way more than most people use, especially for oilseeds.

·

Basic reporting

This paper describes a method of fatty acid quantification by UPLC/MS on rapeseed and sunflower seeds. This method was previously successfully used to quantify fatty acid in different type of organisms. The authors compare the FA profile of 50 lines of each plant species analyzed by the UPLC/MS method or by GC-FID. They conclude that UPLC/MS is more sensitive and detect more FA molecules.

Experimental design

Major remark:
The methods of lipid extraction from the seed material are totally different in the two methods: in the GC-FID method, extraction is done only with hexane and will be therefore specific for highly apolar lipids such as triacylglycerols, whereas in the UPLC/MS method, the extraction is done with the MTBE method that will extract triacylglycerols but also polar lipids such as phospholipids. Therefore, it is not surprising to find more diverse FA molecules with the UPLC/MS method.
There is no comparison of recovery or quantification, only fatty acid distribution… When the number of detected fatty acid changes, the percentage also change… Maybe a comparison by quantity of fatty acid per seed will help to really compare the two methods.
Minor remark:
Is the method of fatty acid hydrolysis specific enough? Normally the use of methanol and KOH produces also transesterification and therefore FAMEs.
Where does the odd chain fatty acid come from?
A lot of details are missing for the GC-FID analysis procedure such as the column length that can explain why the very long chain fatty acids are not detected.

Validity of the findings

No comment

---

## Round 0.2 · Minor Revisions

· Academic Editor

Minor Revisions

Your manuscript was carefully revised. May I ask you to revise your manuscript according to the remaining suggestions of reviewer 2 once more? In addition, I could not see the legends for supplemental figures and tables. Could you please add them to your revision as well?

·

Basic reporting

Well presented manuscript and sufficient literature have provided.No further comments.

Experimental design

No comment.

Validity of the findings

No comment

Additional comments

Well presented methodology with a novel approach using LCMS for FA analysis.

Reviewer 2 ·

Basic reporting

It seems to read better than previously, but there are still some minor issues:

287-288: Sperman should be Spearman (multiple places)
214: rt should be retention time
157: Daltonik should be Daltonic (unless this is a European spelling)
131: Concentrator should not be capitalized.

Supplemental Tables seem to be called 2 and 3 with no 1. I still could not see the legends for supplemental figures and tables.

Experimental design

No comment

Validity of the findings

Table 1A: It seems like it would be hard to detect 3 or 5 ng/ml when these values are only 3-5% more than the y-intercept (and it’s out of the linear range). Similarly, how can you quantify things below the linear range? I understand that you defined the LOD and LOQ based on the signal/noise ratio, but shouldn't at least the LOQ be in the linear range?

·

Basic reporting

The authors corrected well their manuscript, describing through the text that the two methods differ not only by the detection method (GC-FID vs LC-MS) but also by the extraction. Therefore the manuscript is accurate.

Experimental design

No comment

Validity of the findings

No comment

---

## Round 0.3 · accepted · Accept

· Academic Editor

Accept

Thank you for carefully revising your manuscript.

#